# Ipsilateral internal carotid artery web and acute ischemic stroke: A cohort study, systematic review and meta-analysis

Brian Mac Grory[1]*, Erez Nossek[2], Michael E. Reznik[3], Matthew Schrag[4],
Mahesh Jayaraman[3,5], Ryan McTaggart[3,5], Adam de Havenon[6], Shadi Yaghi[3],
Wuwei Feng[1], Karen Furie[3], Anusha Boyanpally[7]

1 Division of Vascular Neurology, Department of Neurology, Duke University School of Medicine, Durham, North Carolina, United States of America, 2 Division of Vascular Neurosurgery, Department of Neurosurgery, New York University School of Medicine, New York City, New York, United States of America, 3 Division of Vascular Neurology, Department of Neurology, Brown University, Providence, Rhode Island, United States of America, 4 Division of Vascular Neurology, Department of Neurology, Vanderbilt University School of Medicine, Nashville, Tennessee, United States of America, 5 Division of Neuroradiology, Department of Radiology, Brown University, Providence, Rhode Island, United States of America, 6 Division of Vascular Neurology, Department of Neurology, University of Utah School of Medicine, Salt Lake City, Utah, United States of America, 7 Division of Vascular Neurology, Department of Neurology, Vidant Medical Center, Greenville, North Carolina, United States of America

* brian.macgrory@duke.edu

**Data Availability Statement:** All relevant data are within the manuscript and its Supporting Information files.

## Abstract

### Introduction

The carotid web is a compelling potential mechanism of embolic ischemic stroke. In this study, we aim to determine the prevalence of ipsilateral carotid web in a cohort of ischemic stroke patients and to perform a systematic review and meta-analysis of similar cohorts.

### Patients & methods

We performed a retrospective, observational, cohort study of acute ischemic stroke patients admitted to a comprehensive stroke center from June 2012 to September 2017. Carotid web was defined on computed tomography angiography (CTA) as a thin shelf of non-calcified tissue immediately distal to the carotid bifurcation. We described the prevalence of carotid artery webs in our cohort, then performed a systematic review and meta-analysis of similar cohorts in the published literature.

### Results

We identified 1,435 potentially eligible patients of whom 879 met criteria for inclusion in our analysis. An ipsilateral carotid web was detected in 4 out of 879 (0.45%) patients, of which 4/4 (1.6%) were in 244 patients with cryptogenic stroke and 3/4 were in 66 (4.5%) patients <60 years old with cryptogenic stroke. Our systematic review yielded 3,192 patients. On meta-analysis, the pooled prevalence of ipsilateral carotid web in cryptogenic stroke patients <60 was 13% (95% CI: 7%-22%; $I^2$ = 66.1%). The relative risk (RR) of ipsilateral versus contralateral carotid web in all patients was 2.5 (95% CI 1.5–4.2, p = 0.0009)

**Funding:** The authors received no specific funding for this work.

**Competing interests:** The authors have declared that no competing interests exist.

whereas in patients less than 60 with cryptogenic stroke it was 3.0 (95% CI 1.6–5.8, p = 0.0011).

## Discussion

Carotid webs are more common in young patients with cryptogenic stroke than in other stroke subtypes. Future studies concerning the diagnosis and secondary prevention of stroke associated with carotid web should focus on this population.

## Introduction

One third of all ischemic strokes have no known cause [1] and there is a fundamental gap in knowledge about the mechanisms of these "cryptogenic" strokes. The carotid web is an intra-luminal projection of hyperplastic intima arising from the carotid artery bulb which causes blood stagnation with the potential for distal embolization [2]. It has been theorized as a cause of ischemic stroke that may be associated with a high risk of recurrent stroke across multiple observational studies [3]. We sought to determine the prevalence of carotid webs in a cohort of patients presenting to a comprehensive stroke center and to combine this with similar studies in a meta-analysis. We hypothesized that ipsilateral carotid web is more common in crypto-genic stroke than in strokes with a known etiology.

## Methods

### Patient population

This study utilized a retrospective, observational, cohort design. We leveraged a large, institutional quality improvement database containing data from consecutive, well-phenotyped ischemic stroke patients presenting to Rhode Island Hospital between June 2012 and September 2017. Data were aggregated through RedCap (Vanderbilt University, Nashville, Tennessee) and this study was approved by the Rhode Island Hospital Institutional Review Board (IRB #1095514–14). The requirement for written, informed consent was waived by the IRB.

### Baseline data acquisition

We enrolled patients who had evidence of acute anterior circulation infarction on neuroimaging (CT or MRI), and cervical vessel imaging with computed tomography angiography (CTA). Adjudication of stroke etiology was performed by a research associate then re-adjudicated by an attending vascular neurologist (BMG) according to the TOAST classification [4]. Within the subgroup of patients with cryptogenic stroke, we then applied separate criteria [5] to classify them as either "ESUS" or "other cryptogenic stroke" (a stroke with two or more competing mechanisms or an incomplete workup).

### Imaging analysis

The carotid web was defined as a shelf-like projection in to the lumen of the proximal internal carotid artery, best visualized on sagittal imaging and corresponding to a septum on axial imaging in the absence of calcification or evidence of arterial dissection according to the method of Choi et al. [6]. After a period of training, each case was adjudicated by one of two investigators (BMG or AB) and ambiguous cases resolved by consensus and consultation with

a third adjudicator. They were explicitly distinguished from other lesions that can mimic their radiographic appearance, including non-calcified atherosclerosis, arterial dissection, and intra-luminal thrombus [6]. Additional consideration was taken not to mislabel "small pro-truding lesions" (SPLs) as carotid webs. These lesions have the same appearance as carotid web on sagittal or sagittal-oblique imaging but do not have evidence of a septum on axial imaging [6].

### Systematic review

A systematic literature review was performed according to the methodology described in **S1 Method**.

### Statistical analysis

Clinical characteristics of the study cohort were presented using descriptive statistics. Owing to the small number of patients with ipsilateral carotid web, inferential statistical analysis was not performed. After meta-analysis, the combined prevalence rate of ipsilateral carotid web in other cohorts of young patients with cryptogenic stroke was estimated using a random effects model. We meta-analyzed relative risk of ipsilateral versus contralateral carotid web from all published cohorts of consecutive stroke patients in whom carotid web presence/absence was adjudicated and performed sensitivity analyses in which we included only patients with cryp-togenic stroke and patients <60 with cryptogenic stroke. Statistical analysis was performed using R software v3.5.1 (R Foundation for Statistical Computing) and meta-analysis performed using the R *meta* package.

## Results

### Patient demographics

There were 1,435 patients in our institutional database during the study period, of whom 879 met criteria for inclusion in our final analysis. The flow chart of patient selection is demon-strated in S1 Fig. Within this cohort, 265 patients (30.1%) had cryptogenic stroke of which 244/265 (92.1%) met criteria for ESUS. Key demographic and clinical characteristics of patients within our cohort are outlined in Table 1.

**Table 1. Key clinical characteristics of patients in the study sample.**

|  | All patients (N = 879) | ESUS (n = 244) | Non-ESUS (n = 635) |
|---|---|---|---|
| Age (mean±SD) | 72.2±14.9 | 68.7±15.2 | 73.5±14.7 |
| Sex (female) | 443 (50.5%) | 121 (49.6%) | 322 (50.7%) |
| Black | 58 (6.6%) | 22 (9%) | 36 (5.7%) |
| Hypertension | 592 (67.3%) | 159 (65.2%) | 433 (68.2%) |
| Hyperlipidemia | 375 (42.7%) | 105 (43%) | 270 (42.5%) |
| Diabetes | 201 (22.9%) | 70 (28.7%) | 131 (20.6%) |
| Coronary Artery Disease | 171 (19.5%) | 46 (18.9%) | 125 (19.7%) |
| Atrial Fibrillation | 276 (31.4%) | 0 (0%) | 274 (43.1%) |
| Smoking | 144 (16.4%) | 37 (15.2%) | 107 (16.9) |
| Admission NIHSS (mean±SD) | 12.9±8.8 | 10.7±8.6 | 13.8±8.7 |
| Ipsilateral carotid web | 4 (0.46%) | 4 (1.6%) | 0 (0%) |
| Contralateral carotid web | 3 (0.34%) | 1 (0.4%) | 2 (0.3%) |

## Prevalence and imaging features of carotid webs

Carotid webs in either carotid artery were detected in 7 out of 879 (0.8%) patients and ipsilateral carotid webs in 4 out of 879 (0.45%). Of the 4 symptomatic carotid webs, all were detected in patients with ESUS. Illustrative imaging examples from our cohort are presented in Fig 1. Three patients with symptomatic carotid web were Black and one was White-Hispanic. Three patients had contralateral carotid webs and were older than the patients with ipsilateral carotid webs (61.6±15 vs. 50.3±16.1). No bilateral webs were identified in our study. None of the identified webs caused flow-limiting stenosis. Carotid webs were present in 4.5% of patients aged <60 with ESUS. In patients with ESUS, SPLs were present in equal measure both ipsilateral and contralateral to stroke– 2.1% of patients.

## Systematic review and meta-analysis

The PRISMA [7] diagram of study selection is included as S2 Fig. In addition to the present study, 9 studies [6, 8–15] were identified with a combined total of 3,192patients of whom 1,127 patients had cryptogenic stroke. We identified 4 studies [8–10, 12] in which we could derive

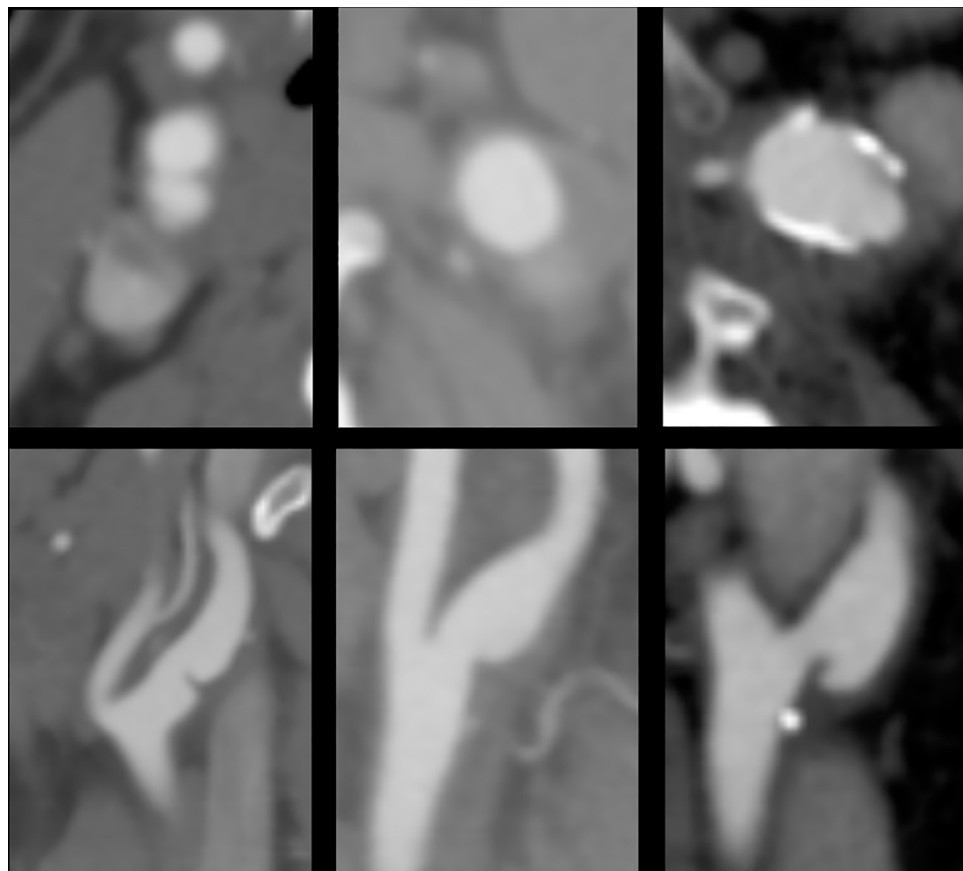

**Fig 1. Representative examples of lesions observed in the proximal internal carotid artery within our cohort.**
Panel A: Internal carotid artery web visible as a protruding lesion in to the lumen of the internal carotid artery seen as a septum on axial imaging (upper panel) and emanating from the posterior wall on oblique sagittal imaging (lower panel). Panel B: Small protruding lesion not visible on axial imaging (upper panel) but sharing imaging features with the carotid web on sagittal oblique imaging (lower panel). Panel C: Atherosclerotic plaque mimicking the conformation of carotid artery web on sagittal oblique imaging (lower panel) but with evidence of eccentric calcified atherosclerosis on axial imaging (upper panel).

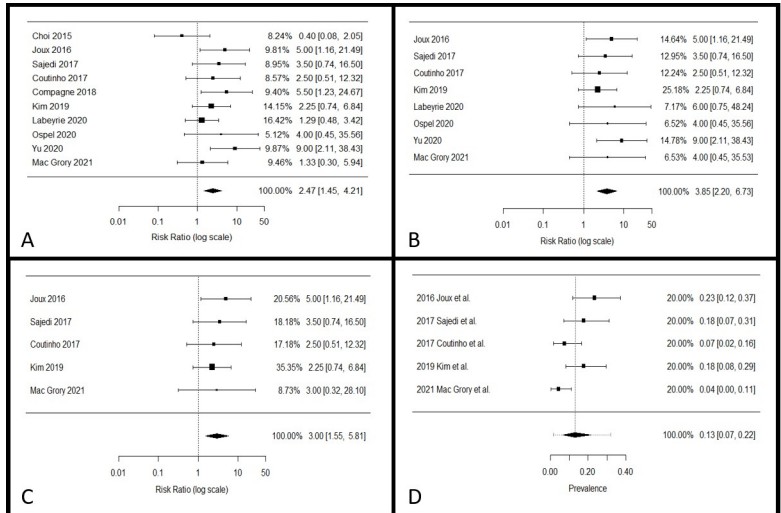

**Fig 2. Meta-analysis of carotid web in ischemic stroke cohorts.** Panel A: Relative risk of ipsilateral versus contralateral carotid web in patients with stroke All studies. ($I^2$ = 27.71%, p = 0.0009). Panel B: Cryptogenic stroke only. ($I^2$ = 0%, p = 0.0001). Panel C: Cryptogenic stroke in patients less than 60 years old. ($I^2$ = 0%, p = 0.001). Panel D: Pooled prevalence of ipsilateral carotid web in young (<60) patients with cryptogenic stroke combined via a random effects model. ($I^2$ = 65.42%). Please note that "Mac Grory 2021" refers to the present study.

data on patients <60 with cryptogenic stroke, a total of 332 patients. The other studies contained a higher proportion of patients of Black patients than our study (ranging from 54.5% [9] to 100% [8]). The relative risk of carotid web ipsilateral versus contralateral to ischemic stroke was 2.5 (95%: 1.5–4.2, p<0.01) in all patients with acute ischemic stroke and 3.0 (95% CI: 1.6–5.8, p<0.01) in patient less than 60 with cryptogenic stroke (Fig 2A–2C). The pooled prevalence of ipsilateral carotid web in patients <60 with cryptogenic stroke was 13% (95% CI: 7%-22%; $I^2$ = 66.1%) (Fig 2D).

## Discussion

In this study, we report the prevalence of carotid webs in a large cohort of consecutive ischemic stroke patients from an institutional registry. This study describes a low incidence of carotid web in an undifferentiated stroke population. Although rare in the stroke population at large, ipsilateral carotid webs were present in 4.6% of young patients with ESUS. Our findings are consistent with prior studies describing a low prevalence of carotid webs in the stroke population at-large but a higher prevalence in young patients with cryptogenic stroke particularly in samples with a high proportion of non-white subjects. Three of four patients with symptomatic carotid web in our study were Black and this is in keeping with trends seen in the literature to-date [3, 16]. On meta-analysis, the relative risk of carotid web ipsilateral versus contralateral to stroke was 3.0 in patients less than 60 with cryptogenic stroke. The lower prevalence of carotid web in our study may reflect the lower number of Black patients in our sample compared with other studies, and our results help further delineate potential ethnic and racial differences in prevalence.

Carotid web is an important potential stroke mechanism as, despite its rarity, it appears from multiple observational studies to be associated with a high risk of recurrent stroke [3]. Carotid webs further share some features of the demographics of FMD particularly the increased prevalence among young women, however, the reproducibly increased prevalence among Black patients is distinctive of carotid webs [17, 18]. FMD of the carotid is associated

with a fairly lower stroke recurrence risk when treated with anti-platelet agents. By contrast, the high recurrence risk associated with carotid web means its early recognition and treatment is crucial to reduce the risk of further stroke.

Our study benefited from a large cohort of patients from a comprehensive stroke center in whom stroke etiology was adjudicated by two separate investigators. Our findings should be considered in terms of a number of limitations: 1) we did not have pathological confirmation of intimal FMD, which is definitively diagnostic of carotid web [19]; 2) within our meta-analysis, the estimated pooled prevalence of 13% of patients <60 with cryptogenic stroke may be an artificially high estimate in part mediated by publication bias; 3) only one other study [14] in the systematic review classified stroke as ESUS and thus there is a small limitation introduced through combining studies with different definitions of cryptogenic stroke and; 4) because our study sample is not population-based, we cannot draw inferences on the prevalence of carotid web in the population at large.

## Conclusions

Carotid web may be detected in nearly 5% of younger patients with cryptogenic stroke, and the higher rate of carotid webs ipsilateral to ischemic strokes may further implicate their involvement as a potential stroke mechanism.

## Supporting information

**S1 Checklist. Preferred reporting items for systematic reviews and meta-analyses checklist.**
(DOCX)

**S1 Fig. Flowsheet of patients included in our analysis.**
(DOCX)

**S2 Fig. PRISMA diagram of study selection.**
(DOCX)

**S1 Methods. Supplementary methodology for systematic review and meta-analysis.**
(DOCX)

**S1 Dataset.**
(XLSX)

## Author Contributions

**Conceptualization:** Brian Mac Grory, Erez Nossek, Matthew Schrag, Mahesh Jayaraman, Ryan McTaggart, Shadi Yaghi, Wuwei Feng, Karen Furie.

**Data curation:** Brian Mac Grory, Erez Nossek, Michael E. Reznik, Shadi Yaghi, Anusha Boyanpally.

**Formal analysis:** Brian Mac Grory, Adam de Havenon, Shadi Yaghi, Anusha Boyanpally.

**Investigation:** Brian Mac Grory, Anusha Boyanpally.

**Methodology:** Brian Mac Grory.

**Project administration:** Brian Mac Grory.

**Writing – original draft:** Brian Mac Grory.

**Writing – review & editing:** Brian Mac Grory, Erez Nossek, Michael E. Reznik, Matthew Schrag, Mahesh Jayaraman, Ryan McTaggart, Adam de Havenon, Shadi Yaghi, Wuwei Feng, Karen Furie, Anusha Boyanpally.

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
