## [Decision Letter · Decision Letter 0]

31 Aug 2021

PONE-D-21-21566

Ipsilateral Internal Carotid Artery Web and Acute Ischemic Stroke – A Cohort Study, Systematic Review and Meta-Analysis

PLOS ONE

Dear Dr. Mac Grory,

Thank you for submitting your manuscript to PLOS ONE. After careful consideration, we feel that it has merit but does not fully meet PLOS ONE’s publication criteria as it currently stands. Therefore, we invite you to submit a revised version of the manuscript that addresses the points raised during the review process.

We look forward to receiving your revised manuscript.

Kind regards,

Aristeidis H. Katsanos, MD, PhD

Academic Editor

PLOS ONE

Journal Requirements:

Reviewers' comments:

Reviewer's Responses to Questions

**Comments to the Author**

1. Is the manuscript technically sound, and do the data support the conclusions?

Reviewer #1: Yes

Reviewer #2: Partly

2. Has the statistical analysis been performed appropriately and rigorously? 

Reviewer #1: Yes

Reviewer #2: Yes

3. Have the authors made all data underlying the findings in their manuscript fully available?

Reviewer #1: Yes

Reviewer #2: Yes

4. Is the manuscript presented in an intelligible fashion and written in standard English?

Reviewer #1: Yes

Reviewer #2: Yes

5. Review Comments to the Author

Reviewer #1: The manuscript by Mac Grory et al describes an observational study on patients with ischemic stroke and carotid webs as well as a systematic review and a meta-analysis. They found that an ipsilateral carotid web occurs in 0.45% of their cohort and in 4.5% in younger patients with a cryptogenic stroke. The pooled prevalence of an ipsilateral carotid web was 13%. The potential role of carotid webs in cryptogenic stroke is interesting. I have one comment:

The pooled prevalence of 13% of ipsilateral carotid web in younger patients with cryptogenic stroke seems rather high. Could this estimation be biased?

Reviewer #2: The present manuscript investigates the prevalence of carotid webs in stroke patients.

There are some ‘typos’ and inconsistencies in the text (for instance the phrase: Carotid web is an important possible stroke mechanism), please review for potential corrections.

Please also check and provide feedback for the following:

The study by Sajedy 2019 reports on the incidence of carotid web among consecutive CTA neck studies, not stroke patients, please explain how this may serve to calculate risk ratios for stroke. Moreover, Ospel et al reports mainly on carotid plaques, not webs. Alternatively, you may remove both studies from the analysis.

The studies by Yo, and Labeyrie et al, which are used in the meta-analysis are not referenced at al. Does Mc Grory et al 2020, reported in the meta-analysis refer to the present study or just to the review referenced in the Supl?

6. PLOS authors have the option to publish the peer review history of their article (what does this mean?). If published, this will include your full peer review and any attached files.

Reviewer #1: No

Reviewer #2: No

---

## [Author Response · Author response to Decision Letter 0]

4 Sep 2021

RESPONSE TO REVIEWERS & EDITORIAL BOARD

We wish to thank the reviewers and the members of the editorial board for the time they have taken in the thoughtful review of our manuscript. We feel that our work has been greatly enriched by the incorporation of these comments. Please find our point-by-point responses below.

Editorial Board:

Thank you for this comment. We have updated the manuscript to adhere with PLOS ONE’s style requirements, including those regarding file naming. 

The following changes have been made.

-Section headers have been changed to bold type, 18pt font.

-Section headers have been changed to sentence case.

-The style of figure callouts has been amended to comply with journal style.

-Figure captions have been added directly after the paragraph in which they are first cited.

-Figure titles now are included in bold type.

-Figures are uploaded separately as individual files.

-Subsections of major sections have been changed to bold type, 16 pt font.

-The table has been included directly after the paragraph in which it is first cited.

-References are cited in square brackets. 

-Supplemental figures have been referenced as S1 Fig and S2 Fig.

-Acknowledgments section has been added. 

-Supporting information captions are included at the end of the manuscript under a Level 1 heading. 

-Supporting information files have been uploaded individually. 

-The title and affiliations page has been modified to comply with journal style.

2. We note that you have indicated that data from this study are available upon request. PLOS only allows data to be available upon request if there are legal or ethical restrictions on sharing data publicly… We will update your Data Availability statement on your behalf to reflect the information you provide.

-Thank you. We apologize for this oversight. We have uploaded an accompanying file entitled “Anonymized data set” as supporting information. This has also been mentioned in the author cover letter.

3. Please include captions for your Supporting Information files at the end of your manuscript, and update any in-text citations to match accordingly. 

-We have now included captions for our Supporting Information files at the end of the manuscript.

Reviewer #1: 

The manuscript by Mac Grory et al describes an observational study on patients with ischemic stroke and carotid webs as well as a systematic review and a meta-analysis. They found that an ipsilateral carotid web occurs in 0.45% of their cohort and in 4.5% in younger patients with a cryptogenic stroke. The pooled prevalence of an ipsilateral carotid web was 13%. The potential role of carotid webs in cryptogenic stroke is interesting. I have one comment:

The pooled prevalence of 13% of ipsilateral carotid web in younger patients with cryptogenic stroke seems rather high. Could this estimation be biased?

-Thank you for this comment. This 13% estimate may certainly be a high estimate and may be in part mediated by publication bias. The confidence intervals are wide and there is a high degree of heterogeneity between studies. We have added the following limitation to the “Discussion” section:

“Our study benefited from a large cohort of patients from a comprehensive stroke center in whom stroke etiology was adjudicated by two separate investigators. Our findings should be considered in terms of a number of limitations: 1) we did not have pathological confirmation of intimal FMD, which is definitively diagnostic of carotid web(15); 2) within our meta-analysis, the estimated pooled prevalence of 13% of patients <60 with cryptogenic stroke may be an artificially high estimate in part mediated by publication bias 3) only one other study(16) in the systematic review classified stroke as ESUS and thus there is a small limitation introduced through combining studies with different definitions of cryptogenic stroke and; 4) because our study sample is not population-based, we cannot draw inferences on the prevalence of carotid web in the population at large.”

Reviewer #2: 

The present manuscript investigates the prevalence of carotid webs in stroke patients.

There are some ‘typos’ and inconsistencies in the text (for instance the phrase: Carotid web is an important possible stroke mechanism), please review for potential corrections.

-Thank you very much for this comment. We have corrected the following typos and inconsistencies:

1. “In this study, we aim to determine the prevalence of ipsilateral carotid webs in a cohort of ischemic stroke patients and to perform a systematic review and meta-analysis of similar cohorts.”

2. “Carotid web was defined on computed tomography angiography (CTA) as a thin shelf of non-calcified tissue immediately distal to the carotid bifurcation.”

3. “Carotid web is an important potential stoke mechanism…”

4. “The lower prevalence of carotid web in our study may be a reflection of reflect the…”

Please also check and provide feedback for the following:

The study by Sajedy 2019 reports on the incidence of carotid web among consecutive CTA neck studies, not stroke patients, please explain how this may serve to calculate risk ratios for stroke. 

-The authors greatly appreciate this comment. As this was study was a study of consecutive CT angiograms of the neck and not of patients with stroke, its inclusion was erroneous. We have removed it from the current analysis and updated Figure 2A, the abstract, results and PRISMA diagram to reflect this. The removal of this study from the analysis of relative risk of ipsilateral versus contralateral carotid web from 2.6 (95% CI:1.6-4.3) to 2.5 (95% CI:1.5-4.2).

Moreover, Ospel et al reports mainly on carotid plaques, not webs. Alternatively, you may remove both studies from the analysis.

-Thank you for this comment. Although the study of Ospel et al was primarily concerned with non-stenotic carotid plaques, they did report on consecutive stroke patients from a prospective study with CTAs of the neck performed. They provided sufficient granularity in their reported results to allow us to determine the number of carotid webs. The numbers abstracted from this paper refer to carotid web specifically and not atherosclerotic plaques. 

The studies by Yo, and Labeyrie et al, which are used in the meta-analysis are not referenced at al. Does Mc Grory et al 2020, reported in the meta-analysis refer to the present study or just to the review referenced in the Supl?

-Thank you very much for this comment. We have added in references to Yu et al. and Labeyrie et al. in the section under Results: Systematic review and meta-analysis. Mac Grory et al. refers to the present cohort and not the prior review paper. We have added the following clarification to the study legend:

“Please note that “Mac Grory 2021” refers to the present study.”

---

## [Decision Letter · Decision Letter 1]

8 Sep 2021

Ipsilateral Internal Carotid Artery Web and Acute Ischemic Stroke – A Cohort Study, Systematic Review and Meta-Analysis

PONE-D-21-21566R1

Dear Dr. Mac Grory,

We’re pleased to inform you that your manuscript has been judged scientifically suitable for publication and will be formally accepted for publication once it meets all outstanding technical requirements.

Kind regards,

Aristeidis H. Katsanos, MD, PhD

Academic Editor

PLOS ONE

Additional Editor Comments (optional):

Reviewers' comments:

Reviewer's Responses to Questions

**Comments to the Author**

1. If the authors have adequately addressed your comments raised in a previous round of review and you feel that this manuscript is now acceptable for publication, you may indicate that here to bypass the “Comments to the Author” section, enter your conflict of interest statement in the “Confidential to Editor” section, and submit your "Accept" recommendation.

Reviewer #1: All comments have been addressed

Reviewer #2: All comments have been addressed

2. Is the manuscript technically sound, and do the data support the conclusions?

Reviewer #1: Yes

Reviewer #2: Yes

3. Has the statistical analysis been performed appropriately and rigorously? 

Reviewer #1: Yes

Reviewer #2: Yes

4. Have the authors made all data underlying the findings in their manuscript fully available?

Reviewer #1: Yes

Reviewer #2: Yes

5. Is the manuscript presented in an intelligible fashion and written in standard English?

Reviewer #1: Yes

Reviewer #2: Yes

6. Review Comments to the Author

Reviewer #1: I have no further comments for the authors regarding this manuscript. The review questions have been addressed sufficiently.

Reviewer #2: (No Response)

7. PLOS authors have the option to publish the peer review history of their article (what does this mean?). If published, this will include your full peer review and any attached files.

Reviewer #1: No

Reviewer #2: **Yes: **Odysseas Kargiotis

---

## [Editor Report · Acceptance letter]

10 Sep 2021

PONE-D-21-21566R1 

Ipsilateral internal carotid artery web and acute ischemic stroke: A cohort study, systematic review and meta-analysis 

Dear Dr. Mac Grory:

I'm pleased to inform you that your manuscript has been deemed suitable for publication in PLOS ONE. Congratulations! Your manuscript is now with our production department. 

Kind regards, 

on behalf of

Dr. Aristeidis H. Katsanos 

Academic Editor

PLOS ONE